# OpenReview forum: "Humans overrely on overconfident language models, across languages"
_colmweb.org/COLM/2025/Conference — COLM 2025_

### Official Review · Reviewer_R4P6 · 2025-05-08

**Rating:** 7
**Confidence:** 4
**Ethics Flag:** 1

**Summary:**

This paper studies LLM overconfidence across multiple languages, and reveals that human reliance on these expressions varies by language, e.g., with Japanese speakers showing greater reliance on uncertainty markers than English speakers reading the same content.

**Questions To Authors:**

- Why did you choose to focus only on MMLU (a multiple-choice dataset) for your evaluation? This seems like a limitation - including at least 2-3 additional benchmark datasets could strengthen your claims about overconfidence across languages.

- How did you make sure that the translations from Google Translate API for languages other than Japanese (i.e., French, German, and Mandarin) were accurate? Since JMMLU was machine-translated and hand-checked, did you (or why did you not) use similar quality control to the other languages?

- The inter-annotator agreement across languages appears moderate at best, particularly for Mandarin (0.26). I do see that you've added some insight into this via a footnote, but do you believe this level of agreement is sufficient for drawing strong conclusions about cross-linguistic patterns (especially for Mandarin)? What could you do to improve agreement?

- Your classifier achieved ~78% accuracy on a held-out test set. What threshold of accuracy did you consider sufficient for reliable annotation, and why? Did you measure inter-annotator agreement between the model "annotators"? I think low agreement could suggest the classifier is not reliable enough.

- In your reliance experiments, bilingual participants completed the task twice (once in each language). Given that the order of languages was randomized, how did you account for potential learning effects? No matter which language comes first, wouldn't participants' behavior naturally differ the second time they do the task?

- If I'm not mistaken (I'm hedging here :-)!), the findings on reliance patterns use bilingual (via English) speakers. To what extent could these results be generalized to monolingual speakers in these countries (who represent I'd imagine the majority of the speakers), who may have different interpretations of certainty markers?

- I'm curious about the omission (as far as I looked) of overconfidence, reliance, and overreliance risk results for few-shot prompted models (basically, what you did for Table 3). Why weren't these included, and is it that these don't provide additional insights that Table 3 does?

- You note that "while all three of these models are designed to be multilingual, the Llama models do not explicitly support Japanese or Mandarin (they do, however, support French and German)." How might this differential language support affect your comparison with GPT-4o, which supports all the tested languages?

**Reasons To Accept:**

- The study addresses a relatively understudied area, and the authors do a terrific job of situating it within prior research. The findings they present had clear takeaways across multiple domains: model behavior risks (AI safety), cross-linguistic differences (computational linguistics), and bilingual user interactions (cross-cultural pragmatics). It made the paper very interdisciplinary and rich!

- Very well written, and I found the sections to transition seemlessly, even between subsections of the related work. The entire paper read like a coherent and focused passage, and I commend the authors for it!

- To examine model behaviour (via overconfidence) accross different languages was already novel and interesting, but the authors drove home the practical implications of this  by bridging that with human reliance, rather than examining these aspects in isolation.

**Reasons To Reject:**

- The study's reliance on a single dataset (MMLU) limits the generalizability of the findings across different types of reasoning tasks and domains.
- Some aspects of the methodology raises concerns about quality control, particularly regarding translation accuracy and inter-annotator agreement in certain languages.
- The study design for measuring human reliance has some limitations that may affect ecological validity, including the use of bilingual rather than monolingual participants as well as potential order effects.

---

> ### Author Response · Authors · 2025-05-31
>
> Thank you very much for the insightful comments! We address your questions here:
>
> 1. **Why only MMLU?** We chose to focus on MMLU as it pulls questions from a wide set of knowledge domains (i.e. multiple subjects) and is somewhat challenging for models, while still making it straightforward to elicit epistemic markers and measure accuracy. Since MMLU includes a diverse variety of domains (e.g. math, history, ethics, science), we believe that our results are indeed broadly generalizable, at least across domains.
> 2. **Google Translate API.** See point 2 of our general response!
> 3. **Inter-annotator reliability.** See point 3 of our general response!
> 4. **Classifier accuracy.** See point 4 of our general response!
> 5. **Learning effects.** This should be mitigated by randomization: since the languages are shown in random order, and we take the average across all experiments, any difference we find should be due to the stimulus, not the ordering. (note that we also randomize questions within each language)
> 6. **Does reliance generalize to monolinguals?** This is a really good question! Due to limitations with data collection (Prolific is an English-language platform), we were only able to focus on bilingual speakers. We agree that future work should study whether this generalizes to monolingual speakers. Our intuition is that it would: the “listener model” employed by each bilingual speaker should be independent for each language, such that an English-French bilingual perceives French epistemic markers in the same way that a French monolingual perceives them. This should be empirically tested, though!
> 7. **Omission of scores for few-shot models.** Could you expand on what you mean here? The numbers in Table 3 (overconfidence and overreliance risk) are computed based on the generations of the few-shot prompted models.
> 8. **Language support for Llama.** This is a good point that we would have liked to spend more time on in the paper. In general: while we don’t observe differences in performance (indeed, Llama gets better MMLU accuracy in Japanese and Mandarin than German), we do notice that the distribution of generations for these languages is qualitatively different between Llama and GPT-4 (Figure 2). As we do not have access to the exact pre-/post-training recipe for Llama or GPT, it is hard to make statements about exactly why this difference exists. However, we hypothesize that it is due to the fact that GPT-4o is likely post-trained for Japanese and Mandarin, while Llama is likely not; Zhou et al. (2024) show that post-training can significantly alter the distribution of epistemic markers. This would explain why the distributions for English, German, and French are somewhat consistent across all models.
>
> Thank you again for taking the time to write your review; we hope we've addressed your concerns.

---

> > ### Comment · Reviewer_R4P6 · 2025-06-03
> > **Response to Author**
> >
> > Thanks a lot for your response! Much of your clarifications and additional results helped.
> >
> > Here are some of my thoughts briefly on specific responses:
> >
> > **MMLU Dataset**:  I do see the explanation about MMLU's domain diversity as reasonable, but still believe including at least one additional dataset type would strengthen claims about generalizability (since it is more than just a matter of domains).
> >
> > **Translation Quality**: I think it's more aproppriate to acknowledge this as a limitation, since I don't believe relying on the *lack* of negative feedback from annotators as a strong validation method.
> >
> > **Inter-annotator Reliability**: The additional analysis was helpeful! Substantially higher kappa scores when removing moderate annotations was interesting.
> >
> > **Learning effects and monolingual generalization**: Both explanations were reasonable and I appreciate the acknowledgement that the latter needs empirical tests.
> >
> > **Few-shot models, Table 3**: Thanks for clarifying! I had at the point forgotten that the results are generated via few-shot prompting (and you had an interesting justification about this in the general response, although I didn't think of the limitation originally), and not 0-shot.
> >
> > **Language Support**: That's an interesting hypothesis, thank you!
> >
> > I have accordingly raised my score since most of my more substantial concerns were addressed with the additional results and clarifications.

---

### Official Review · Reviewer_NTZ4 · 2025-05-08

**Rating:** 7
**Confidence:** 4
**Ethics Flag:** 1

**Summary:**

The authors study how/if LLMs exhibit linguistic (mis)calibration across different languages. They find that there are differences and that speakers of the different languages react differently to signifiers (that either strengthen or weaken a statement.)

I find this paper great but I have a couple questions about some specific design choices and I'm excited to hear the authors' response.

**Questions To Authors:**

# Questions
0. Is the full prompt included in the appendix or is there more to it?
1. Building off the first [reason to reject] above, is there a way to control for this effect? What happens if you only instruct the model to produce calibrated epistemic markers without the rigid fewshot examples? Or, if you add additional text like: If you are confident produce examples like X; If you are uncertain produce examples like Y. Do you have some general response to this line of questions?
2. Does "plain language" fall into the moderate bucket? Is there a reason that you could not add this bucket (w/o weakeners/strengtheners) to most figures and setups?
3. It seems like a lot of the questions are *conditioned on the fact that an epistemic marker is generated...* and if the rate of this happening is relatively rare, as you suggest, and if there may be differences between this occurring with/without explicit prompting then the effects, then the effects shown might not be a strong/clear as the figures make them out to be. Or at least, take for example, this quote from Figure 3: "Generations broadly match linguistic norms from the literature, i.e., in Japanese, models produce more hedges (weakeners) than boosters (strengthers)". This could be true, but what if the ratios would tend to be all very small compared to plain language? Then while still maybe true the interpretation of the results may be different.
4. Why is Chinese not included in some figures? (Perhaps there was a note that I missed.)
5. Figure 5 is interesting: Plain language seems to be the most "relied" on type of expression but the comment/caption is about the strong vs moderate. I think pointing out that plain language is interpreted to be the most "confident" across all languages is worth mentioning and probably elevating into the main body of the paper if you can find room.
6. Do you find your fleiss agreement scores good? What are the agreement scores between the model classifiers? What do the disagreements look like?

# Notes
* 182: "moderate" should be fair to moderate as in the table
* What is the agreement metrics on the classifiers? Is that listed in an appendix?

**Reasons To Accept:**

* I find your figures clear, legible, and easy to understand!
* Your tables are also very clear! They look great and read well
* The research questions and approach are clear and well done

**Reasons To Reject:**

I had one primary concern about the work's methods. There is something artificial about encouraging models to produce the strengtheners/weakeners via a few-shot prompt and then evaluating them.[^1] There seems to be more than one way for a model to interpret that type of prompting. (157-161). Note that the strengtheners/weakeners within the prompt themselves are not calibrated to the model's capabilities so that might introduce a confounding factor. (Perhaps this setup is as close to a natural yet controlled setting that can be achieved but it seems well possible that models may behave differently when they "spontaneously" generate the epistemic markers.)

[^1]:  Evaluating the human responses to model generated epistemic markers seems fine regardless (the second half of the study.)

Some might argue the paper makes limited technical contributions/interest but overall I found the paper's research questions interesting and the approach well done.

---

> ### Author Response · Authors · 2025-05-31
>
> Thank you for taking the time to write such a detailed review! We address your questions here:
>
> 1. **Prompt in the appendix.** The prompt in the appendix is an example of one of the English few-shot prompts we used. We sampled three few-shot prompts in total. Few-shot examples were translated across languages. We will release all prompts with the camera-ready.
> 2. **Eliciting epistemic markers from few-shot prompts.** See point 1 of our general response!
> 3. **Plain language only in reliance study.** Since we prompt models to produce epistemic markers in Section 3, models almost never fail to produce a marker. We include plain language in Section 4’s reliance study as a control. Plain expressions were hand generated.
> 4. **If markers are rare without prompting, does this change our results?** See point 1 of our general response! At a high level, no: overreliance risks are actually even higher for plain expressions than for generations including strengtheners.
> 5. **Why is Chinese not included in reliance figures?** See footnote 6 — because all models generally failed to generate weakeners in Mandarin, we excluded it from the reliance studies.
> 6. **Plain language is most relied on across languages.** We agree that this is an interesting result, and will add it to the main text for the camera ready!
> 7. **Fleiss scores.** See points 3 and 4 of our general response!
>
> Again, thanks so much for the review, and let us know if this addressed your questions!

---

> ### Comment · Reviewer_NTZ4 · 2025-06-03
>
> Thanks for your response!
>
> I am curious about your "Why do we need few-shot prompting to elicit markers? Is this artificial?" response. If plain language is worse than markers and markers do not tend to be generated by models naturally (and if they are generated they are known to be somewhat over-relied on) doesn't that seem like we're investigating the wrong thing here?
>
> There seems to be an argument here that suggests there is not much point in encouraging models to generate these type of markers. Or do you view it as the best of the available bad options?
>
> Could you respond to the following? (Mainly the last part, that the markers within the prompts are not necessarily calibrated to the model being tested.)
>
> > I had one primary concern about the work's methods. There is something artificial about encouraging models to produce the strengtheners/weakeners via a few-shot prompt and then evaluating them. There seems to be more than one way for a model to interpret that type of prompting. (157-161). Note that the strengtheners/weakeners within the prompt themselves are not calibrated to the model's capabilities so that might introduce a confounding factor.

---

> > ### Author Response · Authors · 2025-06-08
> >
> > Thanks!
> >
> > Our general way of thinking about plain language is: plain language is worse for reliance (i.e. humans tend to rely on them heavily, and there is no variation), but also what models tend to produce naturally. Human reliance is much better calibrated with graded epistemic markers (i.e. reliance will not be uniformly high), and thus we should try to elicit epistemic markers from models.
> >
> > For your second comment - we agree that this could introduce a confound, but our setup mitigates it as much as possible. In our initial experiments, we tried various different setups---all correct, all incorrect, perfect calibration (i.e. strengtheners correct, weakeners incorrect), etc.---and found that this made little difference with regards to distribution/overconfidence. The setup we ended up going with (all correct) was the best middle ground in terms of eliminating potential confounds: the prompt does not encode a correlation between strengtheners and correctness (since correct responses are equally likely to use weakeners), so since we're focusing on the overconfidence rate in particular (i.e. when models use strengtheners for _incorrect_ responses) this should not be affected (i.e. increased) by the prompt.
> >
> > Again, thanks so much - let us know if this addresses your questions!

---

### Official Review · Reviewer_TxkE · 2025-05-12

**Rating:** 7
**Confidence:** 3
**Ethics Flag:** 1

**Summary:**

This paper studies the distribution and effects of epistemic uncertainty markers in model responses, across languages. Firstly, they measure the distribution of uncertainty or certainty markers in a GPT4o and 2 Llama models, across five languages. Secondly, in a trivia quiz, they give humans access to LLM responses to measure their reliance on epistemic markers when deciding how to answer. They do find model over-confidence, and human overreliance, but most notably, they find it differs across linguistic settings, motivating cultural and linguistic context in such evaluations.

**Questions To Authors:**

Refer to the reasons to reject section. Would love to hear the authors thoughts on these critiques.

Also, you discuss over-confidence rates, but not under-confidence rates at which the model gave weak epistemic strength with correct answers (i.e. under-confidence/hedging)?

**Reasons To Accept:**

For me, while the portion of the paper examining the distribution of LLM epistemic markers is less compelling, the second half of this work is extremely interesting, and motivates most of my reasons to advocate for it being accepted. In particular, the papers strengths:

* The authors devise an insightful experimental setup to demonstrate how frequently users might rely on, or be misled by LLMs use of epistemic markers.
* The results illustrate how the strength of epistemic uncertainty differs by language, and people rely on these markers differently by language. They articulate the implications well: “Applying our understanding of how English-speaking users might rely on English markers would have provided us with an inaccurate estimation of overreliance risks of Japanese generations. Our findings show that reliance on the epistemic markers is dependent on the linguistic norms and emphasizes the need to evaluate overreliance in context.” This is a very relevant finding for the community and could inform future work on overreliance.
* They provide interesting metrics to measure meaningful risk, by combining overreliance and overconfidence.

**Reasons To Reject:**

The first section of the paper, and some of the assumptions/limitations of the experimental setup may weaken this work:

* The way the models were prompted was designed to coax epistemic markers in their answers. This study would benefit from understanding the prevalence of these markers without prompts designed to elicit them.
MMLU is US/western-centric in its content. Translation will not solve this. It may have ramifications for the study. Diversifying cultural content may have changed some of the early findings?
* Using the Google Translate API may also affect user confidence in translated languages, if translations are imperfect?

---

> ### Author Response · Authors · 2025-05-31
>
> Thanks for the helpful feedback! We address your questions here:
>
> 1. **Why elicit epistemic markers through few-shot prompts?** See point 1 in our general response!
> 2. **MMLU is US/Western-centric.** We agree that MMLU is quite US/Western-centric! We acted to mitigate this in our experiments by removing culture specific questions. To find these, we took the intersection of MMLU and JMMLU. JMMLU replaces “Western culture” questions with equivalent “Japanese culture” questions. By taking the intersection of these two datasets, we removed (most) culture-subjective questions from the dataset. We will add text in the camera-ready to clarify this.
> 3. **Google Translate API.** See point 2 in our general response!
> 4. **Underconfidence rates.** While we don’t report these numerically in the paper, these are available in Figure 1: underconfidence is the probability of a correct response given a weakener, i.e. the leftmost bar in each facet. We will include a table with these numbers in the appendix of the camera ready! Note though that these are less concerning from a safety perspective: the risks of relying on an underconfident model are much lower than that of an overconfident one.
>
> Thanks again for the insightful comments; hopefully this addresses your concerns!

---

> > ### Comment · Reviewer_TxkE · 2025-06-09
> >
> > Thank you for your thorough responses.
> >
> > I understand the context of the decisions better now, and will keep my score the same at supporting acceptance.

---

### Official Review · Reviewer_EaEa · 2025-05-13

**Rating:** 6
**Confidence:** 4
**Ethics Flag:** 1

**Summary:**

This paper addresses the important and timely issue of model overconfidence in multilingual settings. The authors evaluate three language models from two model families across five languages, analyzing disparities between English and non-English outputs through human evaluations. The paper is well-written and the motivation is clear.

**Questions To Authors:**

Questions:

- [1] How did you define the confidence categories (low, medium, high)?
- [2] Did you assess inter-annotator agreement across different languages?
- [3] Could cultural or linguistic factors influence how confidence is perceived across languages?
- [4] How did you select the target languages, and to what extent does the resource availability (low-resource vs. high-resource) impact your results?
- [5] What are potential strategies to mitigate the overconfidence issue you highlight, and how might this affect downstream tasks?
- [6] Could this overconfidence manifest in harmful ways, especially in multilingual or cross-cultural contexts? Could you elaborate?

Suggestions:

- It would be helpful to include explicit examples in the paper illustrating what low, medium, and high confidence look like across different languages.
- Please consider providing more details on how the prompts were constructed, along with representative examples in the paper.

**Reasons To Accept:**

- The paper tackles a meaningful and underexplored problem in multilingual NLP.
- The authors conduct evaluations across multiple languages, which is resource-intensive.
- The use of human annotators with language proficiency adds credibility to the evaluation.

**Reasons To Reject:**

- The paper falls short in suggesting or exploring solutions to the problem it identifies, how might overconfidence be mitigated?
- The analysis lacks consideration of deeper linguistic or cultural dimensions that may impact expressions of confidence across languages.
- The categorization of confidence levels (low, medium, high) is unclear. The criteria for these categories and how they are defined across languages are not well explained.
- It is also unclear whether the authors measured inter-annotator variance across languages or examined whether confidence judgments differ systematically by language or annotator background.

---

> ### Author Response · Authors · 2025-05-31
>
> Thanks so much for your detailed review and feedback! We address your questions below:
>
> 1. **Definition of confidence categories.** As we mention in the paper [Appendix A.1], confidence categories are assigned based on bucketing a 1-7 Likert scale (1-2 = weak, 3-5 = moderate, 6-7 = strong). We will make sure to clarify this in the camera ready.
> 2. **Inter-annotator agreement across different languages.** Could you expand on what you mean by this?
> 3. **Cultural/linguistic effects on perception.** We agree that cultural and linguistic factors influence the perception of confidence across languages! This is what we explore in Section 4 of the paper, and find that there is indeed cross-linguistic variation in reliance on epistemic markers.
> 4. **Language selection.** We chose the five languages in the paper (English, French, German, Japanese, Mandarin) as they are generally high resource in terms of language modeling, and because of the availability of fluent human participants. There are, of course, resource differences between them, and we do observe that this empirically affects performance: in English, models have higher MMLU accuracy (Table 1) and are generally better at instruction following (Table 5, though note French!). However, this does not impact our high level story, as MMLU accuracy does not vary significantly between languages (always between 75-80%).
> 5. **Strategies to mitigate overconfidence.** We believe that this is an exciting area for future work, but is beyond the scope of our paper. Our goal is to highlight (1) existing overconfidence risks and (2) issues with purely English-based approaches to studying overreliance. Prior work (e.g. Zhou et al. 2023, 2024) has explored strategies for mitigating overconfidence in English (e.g. cleaning pre-training data, re-constructing preference datasets), and future work should explore this in the multilingual setting!
> 6. **Harmful manifestations of overconfidence.** We will add some examples in the introduction of the camera ready! The canonical example from the literature is a situation in which an LLM is deployed in a high-risk setting (e.g. medical, legal), and stakeholders are told confident but false information.
> 7. **Explicit examples of markers.** We include these in Appendix A.3!
> 7. **Explicit examples of prompts.** We include an example in Appendix A.3!

---

### Author Response · Authors · 2025-05-31
**General Response**

Thank you to all reviewers for taking the time to write such detailed reviews! We've added individual responses to concerns from each reviewer. We also noticed that there were a few high-level repeat questions / concerns across reviews; we're including responses to those here.

1. **Why do we need few-shot prompting to elicit markers? Is this artificial?**

We understand the concern that specifically prompting models to generate epistemic markers seems less natural than if they were to generate them naturally. The problem is that Zhou et al. (2024a) find that current models are in general very reluctant to elicit epistemic markers with their responses; nearly 90% of responses do not include verbalized confidence. Yet humans have been shown to overrely on plain statements (“the answer is X”) even more than they do on strengtheners (Zhou et al., 2024b; also see Figure 5 of our paper). In other words, plain statements lead to a high risk of human over-reliance.

Since humans are sensitive to linguistic markers of uncertainty (i.e. people rely less on a model when it hedges) utilizing few-shot prompting to elicit verbalized uncertainty is the correct paradigm to mitigate overreliance risks: future systems/models should express their epistemic states verbally through epistemic markers, since this is the interface humans are most responsive to. We therefore need to understand how current models do so, even if this requires stronger elicitation.

We will make this clearer in the camera ready. We will include the following at the end of paragraph 1 of the introduction:
> Past work has shown that models are reluctant to generate epistemic markers with their responses, but that users are highly overreliant on these “plain” statements. As such, it is important that models are well-calibrated to verbalized confidence, which people are sensitive to.

and will bring some of the data about plain statement reliance rates (from Figure 5 in the appendix) into the main text of the paper, for brief discussion.

2. **Effects of machine translation?**

We agree that machine translation is imperfect, and we will add this as a limitation in the camera-ready! To mitigate this to at least some degree, we allowed Prolific annotators to leave feedback on each generation; for all three machine translated languages, we received no comments. Also note that Google Translate is generally high accuracy, and that all three languages (French, German, Mandarin) are high resource.

3. **Inter-rater reliability scores (Fleiss' kappa)**

Our inter-rater reliability scores are fair-to-moderate, and we agree with reviewers that this is a limitation. We implemented a number of mechanisms to ensure that annotations were high quality (qualifying tasks, competency checks, handwriting checks), but this is an inherent limitation of using crowdworkers for linguistic annotation.

However, it is important to note that our central claims in Section 3 focus on cross-linguistic differences between the distribution of strong and weak markers, not moderate markers. We find that the primary source of confusion for annotators is the moderate case (i.e. strong vs. moderate and weak vs. moderate), and that annotators are reliably good at distinguishing strong and weak markers: when we remove moderate annotations, kappa scores increase significantly (French: 0.65, English: 0.92, German: 0.70, Japanese: 0.95, Mandarin: 0.90). As such, we argue that our core findings still reliably hold. We will include this data in the camera ready.

4. **Classifier accuracy for Llama generations**

Each classifier serves as a proxy for a single human annotator. We benchmarked the “expected accuracy” of a classifier by measuring the average accuracy of each human annotator as a classifier (filtering out their own annotations); across languages, we found an expected ~80% accuracy. We will add this data to the camera-ready!

We also want to emphasize that what is important to measure here is how accurate the classifier is to human labels, not how well the classifiers agree with one another. Indeed, it could be possible for all three classifiers to have low accuracy but high agreement; this is why we do not report kappa scores for classifiers.

Also note that we only used the classifier for annotating Llama generations, not for GPT-4o (for which we used hand-annotated generations).

---

### Decision · Program_Chairs · 2025-07-08

**Decision:**

Accept

**Comment:**

This paper evaluates three language models across five languages, analyzing the differences between English and non-English outputs through human evaluations and model overconfidence, finding that model over-confidence differs across linguistic settings. The paper is clear and well-explained, and the authors engaged extensively with the reviewers' comments.

There are some comments and suggestions from the reviewers that should ideally be addressed in the camera-ready version of the paper:
- Proposing more mitigation strategies for over-confidence (or over-reliance on LLM over-confidence)
- More in-depth explanations of certain methodological aspects, e.g. categorization of confidence levels, how inter-annotator variance across languages is measured and whether confidence judgments differ systematically according to different languages or annotator backgrounds
- Ideally, testing on other datasets (apart from MMLU would be helpful as well).

Overall, it is an interesting paper and strong topic that is of interest to the CoLM community.